# Analysis of the Asymmetry between Both Eyes in Early Diagnosis of Glaucoma Combining Features Extracted from Retinal Images and OCTs into Classification Models

**DOI:** 10.3390/s23104737

**Published:** 2023-05-14

**Authors:** Francisco Rodríguez-Robles, Rafael Verdú-Monedero, Rafael Berenguer-Vidal, Juan Morales-Sánchez, Inmaculada Sellés-Navarro

**Affiliations:** 1Departamento de Tecnologías de la Información y las Comunicaciones, Universidad Politécnica de Cartagena, 30202 Cartagena, Spain; franroroba@gmail.com (F.R.-R.); juan.morales@upct.es (J.M.-S.); 2Departamento de Ciencias Politécnicas, Universidad Católica de Murcia (UCAM), 30107 Guadalupe, Spain; rberenguer@ucam.edu; 3Hospital General Universitario Reina Sofía, 30003 Murcia, Spain; inmasell@um.es

**Keywords:** optical coherence tomography (OCT), retinal fundus images, retinal nerve fiber layer (RNFL), RNFL thickness asymmetry, cup-to-disc ratio (CDR), rim-to-disc ratio (RDR), optic nerve head (ONH), retinal imaging analysis, glaucoma, decision trees, support vector machines

## Abstract

This study aims to analyze the asymmetry between both eyes of the same patient for the early diagnosis of glaucoma. Two imaging modalities, retinal fundus images and optical coherence tomographies (OCTs), have been considered in order to compare their different capabilities for glaucoma detection. From retinal fundus images, the difference between cup/disc ratio and the width of the optic rim has been extracted. Analogously, the thickness of the retinal nerve fiber layer has been measured in spectral-domain optical coherence tomographies. These measurements have been considered as asymmetry characteristics between eyes in the modeling of decision trees and support vector machines for the classification of healthy and glaucoma patients. The main contribution of this work is indeed the use of different classification models with both imaging modalities to jointly exploit the strengths of each of these modalities for the same diagnostic purpose based on the asymmetry characteristics between the eyes of the patient. The results show that the optimized classification models provide better performance with OCT asymmetry features between both eyes (sensitivity 80.9%, specificity 88.2%, precision 66.7%, accuracy 86.5%) than with those extracted from retinographies, although a linear relationship has been found between certain asymmetry features extracted from both imaging modalities. Therefore, the resulting performance of the models based on asymmetry features proves their ability to differentiate healthy from glaucoma patients using those metrics. Models trained from fundus characteristics are a useful option as a glaucoma screening method in the healthy population, although with lower performance than those trained from the thickness of the peripapillary retinal nerve fiber layer. In both imaging modalities, the asymmetry of morphological characteristics can be used as a glaucoma indicator, as detailed in this work.

## 1. Introduction

Glaucoma is a silent optic neuropathy in which the visual field is progressively reduced until total vision loss [1]. It is the second leading cause of blindness in developed countries behind cataracts and the leading cause in cases of irreversible blindness [2] by damaging the optic nerve beyond recovery. The fact that it is initially asymptomatic and usually detected in very advanced stages justifies the importance of early diagnosis [3]. The benefit provided by early diagnosis leads to the development of automatic detection procedures easily transferable to primary care centers.

The main risk factor for the development of glaucomatous neuropathy is an elevated intraocular pressure due to alterations in the balance between the production and drainage of aqueous humor, resulting in an increased intraocular pressure (IOP) [4,5]. The most usual form of the disease is primary open-angle glaucoma (POAG), characterized by the maintenance of an open drainage angle between the cornea and iris, yet with partial obstruction of the trabecular meshwork, resulting in an elevation of intraocular pressure [6].

The diagnosis of glaucoma primarily relies on measuring the IOP through tonometry, examining the visual field through campimetry, computing the relationship between the cup and optic disc diameter in retinographies, and evaluating the peripapillar thickness of the retinal nerve fiber layer (RNFL) in optical coherence tomography (OCT) [1]. This work analyzes the ability for early diagnosis of glaucoma using the asymmetry characteristics between both eyes extracted from these two imaging modalities, retinographies and OCTs, and studying their combined use for the diagnosis of glaucoma.

A retinography is an image of the fundus of the eye [7]. It is a non-invasive, simple, fast, and painless test that allows us to appreciate the optic disc and the optic cup of the optic nerve head (ONH). The edge of the optic disc defines where the neuroretinal rim and the sclera meet, whereas the optic cup is the area of the optic disc that is devoid of nerve fibers, typically seen as a depression in the center of the disc. The thickness of both contours in the ONH provides us with valuable information for glaucoma detection since the disease produces noticeable changes in the optic cup shape [8]. The neuroretinal ring is formed by the axons of the retinal ganglion cells. In glaucoma, the death of these cells leads to axonal loss and thus to an increase in the size of the optic cum. This effect can be studied through the comparison of both morphologies, represented by the cup-to-disc ratio (CDR) [9]. The rim-to-disc ratio (RDR), a parameter that refers to the minimum thickness between the optic cup and the optic disc, has proven to be even better than the CDR in characterizing healthy patients and those with glaucoma [10] and has also been evaluated in this study.

The second imaging modality considered in this work is optical coherence tomography, which provides a cross-sectional image of the retina and the optic nerve obtained through the use of infrared light and interferometry [11]. It allows the study of various ocular diseases by visualizing the anatomy of the RNFL surrounding the optic nerve [12]. In particular, glaucoma produces a narrowing of the nerve fiber layer due to increased IOP [13]. Thus, comparison of the thickness of the RNFL between the two eyes may be also of interest in the diagnosis of glaucoma.

In this study, metrics of inter-eye asymmetry provided by these two imaging modalities are analyzed for early diagnosis of glaucoma. Specifically, the contours of the optic disc and optic cup visualized in retinal fundus images as well as the thickness of the RNFL measured in intrasubject OCTs are evaluated. This work extends and improves the approach followed in [14], where only characteristics based on OCT were used. As we will see in Section 2.4, the diagnosis of glaucoma from a retinal fundus image is expected to be less accurate than with an OCT because the parameters extracted from a retinal fundus image, i.e., the contours of the disc and cup, are more subjective even in segmentations performed by experts [15]. In contrast, the thickness of the RNFL obtained from an OCT can be determined more accurately by either a manual or automatic segmentation [16], which should lead to a greater diagnostic accuracy. The estimated accuracy in the glaucoma diagnosis using these imaging modalities is inversely related to their cost and complexity, with retinographies having a lower cost and increased availability in medical centers compared to OCT devices. This suggests a joint study of both imaging modalities in order to exploit their strengths, namely the lower cost and greater availability in the case of retinographies and the greater diagnostic accuracy in the case of OCTs. The analysis of both imaging modalities together with the use of classification models based on asymmetry features from both eyes are the main novelties of this work.

The rest of the article is organized as follows. In Section 2, we describe the image dataset used in this work detailing the extraction of the images of the patients under study. The asymmetry metrics considered and the machine learning models implemented for this work are also addressed in this section. The results obtained with the proposed approach are described in Section 3, which justifies the use of characteristics based on asymmetry in retinal fundus images and OCT images for the early diagnosis of glaucoma using machine learning models, as discussed in Section 4.

## 2. Materials and Methods

### 2.1. Image Datasets

The database for this study has been developed with the assistance of the Ophthalmology Service at the Hospital General Universitario Reina Sofía (Murcia, Spain). The study has been conducted on 293 patients. It contains retinographies of both eyes of 284 patients and OCTs of 242 patients. All the images have been anonymized in accordance with the guidelines established by the Human Ethics Committee of the hospital.

The retinal fundus images have been acquired using a non-mydriatic Topcon TRC-NW400 retinal camera (Topcon Europe Medical BV, Capelle aan den IJssel, The Netherlands) with a resolution of 1934 × 2576 pixels and stored in DICOM format. Additionally, the contour of the optic cup and the optic disc of each retinal fundus image was delineated by two ophthalmologists. The set of images, the segmentations, and the clinical data are publicly available in the dataset called PAPILA [17]. Further details and information regarding the fundus images can be found on [18].

The second set of images used in this research is composed of 2D peripapillary B-scan optical coherence tomography (OCT) images centered at the optic nerve head. These images were acquired using a Spectralis OCT S2610-CB (Heidelberg Engineering GmbH, Heidelberg, Germany) from October 2018 to November 2020. The OCTs have a resolution of 768 × 496 pixels, a bit depth of 8 bits/pixel in grayscale, and a z-scaling of 3.87 µm/pixel.

The patients have been classified by ophthalmologists as healthy (161), with glaucoma (47), and as suspicious (34) when the presence or absence of the disease could not be determined with certainty. In this work, supervised learning techniques are used and only patients whose eyes are both labeled as healthy or with glaucoma are considered. Table 1 gathers the sex and age range of the healthy and glaucoma patients of the dataset. Suspected patients have been discarded since they do not provide useful information to improve the characterization capabilities of the models.

### 2.2. Division of the Information into Sectors

In order to facilitate the study and extraction of features from retinographies and OCTs, the structures provided by these two imaging modalities are typically divided into six sectors: temporal (T), temporal superior (TS), nasal superior (NS), nasal (N), nasal inferior (NI), and temporal inferior (TI). An additional sector (G) refers to the average of all sectors. Note that sectors T and N cover an angle of ninety degrees, while the other sectors cover an angle of forty-five degrees. Figure 1 depicts the distribution of the sectors for both the right and left eyes. Figure 2 shows the correspondence of the sectors in a retinal fundus image with an OCT.

By combining sectors TI and NI as *I* and TS and NS as *S*, we find a grouping of four sectors (*I*, *S*, *N*, *T*) on which we will study the compliance of the ISNT rule [19], which states that in healthy patients there is generally a consistent order of the rim width by sectors in retinographies, concretely,
(1)I≥S≥N≥T.

This rule has been confirmed by other authors [20], and various studies have found a correlation between non-compliance of this rule and damage to the nerve tissue as a result of glaucoma. However, its reliability increases when combined with other measurements such as the CDR [21]. In this study, the compliance of this rule is examined with the retinographies of the PAPILA dataset, as well as its variants IST, i.e., I≥S≥T, and IS, i.e., I≥S [22], which will also be considered as differentiating characteristics with the machine learning models.

### 2.3. Asymmetry Metrics

In order to provide the input characteristics of the machine learning models, several asymmetry metrics are computed using the information extracted from the retinal fundus images (width of the optic rim, CDR and RDR) and OCT (thickness of the RNFL). These metrics were previously defined in [14] and are based on relative and absolute differences considering different normalizations. As shown and discussed in [14], the values of the metrics without absolute differences follow a Gaussian distribution, both for healthy and glaucoma patients, with zero mean but different variance. By applying the modulus to these measurements, the resulting values follow a mean-normal distribution in which the variance of the first metric is now reflected in the mean, allowing for greater a priori separation between the sets of points of both classes.

The metrics defined in [14] to quantify the asymmetry between the characteristics extracted from both eyes are the following: (2)$$\begin{array}{ccc} \delta_{S,i} & = & w_{S,i}^{r}-w_{S,i}^{l}, \end{array}$$(3)$$\begin{array}{ccc} |\delta_{S,i}| & = & |w_{S,i}^{r}-w_{S,i}^{l}|, \end{array}$$(4)$$\begin{array}{ccc} \Delta_{S,i} & = & \frac{w_{S,i}^{r}-w_{S,i}^{l}}{w_{S,i}^{r}+w_{S,i}^{l}}, \end{array}$$(5)$$\begin{array}{ccc} |\Delta_{S,i}| & = & \frac{|w_{S,i}^{r}-w_{S,i}^{l}|}{w_{S,i}^{r}+w_{S,i}^{l}}, \end{array}$$(6)$$\begin{array}{ccc} {\bar{\Delta }}_{S,i} & = & \frac{w_{S,i}^{r}-w_{S,i}^{l}}{w_{S}^{r}+w_{S}^{l}}, \end{array}$$(7)$$\begin{array}{ccc} {\bar{\bar{\Delta }}}_{S,i} & = & \frac{w_{S,i}^{r}-w_{S,i}^{l}}{w_{G,i}^{r}+w_{G,i}^{l}}, \end{array}$$
where *S* denotes the specific sector (TS, T, TI, NS, N, NI, G) in which the asymmetry is computed, *i* refers to the patient index, and r and l indicate the right eye and the left eye, respectively. The first asymmetry metric defined in Equation (Equation 2) computes the difference in thickness between both eyes in sector *S* for patient *i*. The second metric, Equation (Equation 3), applies the modulus, making the direction in which the subtraction is calculated irrelevant. Note that the metrics defined in Equations (Equation 4) and (Equation 5) provide a difference which is normalized with respect to the sum of the thicknesses of that sector of the patient; in Equation (Equation 6), the normalization is based on the average values of each sector for all patients, and in Equation (Equation 7), the difference is divided by the sum of the global sectors of both eyes of the specific patient. The reader is referred to [14] for further details about the asymmetry metrics.

### 2.4. Extraction of Information from Imaging Modalities

Once the asymmetry metrics are defined, the information used for the calculation of these metrics must be extracted from both imaging modalities. For the OCT, the thickness in each sector is provided by the device, although image processing methods can also be applied; see, e.g., [16,23,24,25].

For the retinographies, although there are many automatic methods in the literature for optic disc and cup segmentation [15,26,27,28], the manual segmentation provided in the PAPILA dataset is used [18]. The CDR, RDR, and thickness measurement have been deployed using the ophthalmologist contours as ground truth. Firstly, these points were fitted with ellipses in order to decrease the variance between the points and the noise resulting from manual marking. In order to establish a similarity with the OCT results, the thickness values have been multiplied by a scaling factor. This results in the mean thickness in the general sector (G) obtained from retinography being as close as possible to that obtained from OCTs. Figure 3 shows a retinal fundus image of the right eye of a healthy patient with all the information extracted from the contours together with the location of the RDR. The image on the right shows in blue the contour of the optic disc, in green the contour of the excavation, and in red the difference between them, in addition to a horizontal line with the average value of each of these curves. The sectors are distributed, as typical OCTs are, with the corresponding labeling. The location of the value with the minimum thickness or RDR is plotted in purple.

## 3. Results

### 3.1. Compliance of the ISNT Rule and Its Variants

Given that the PAPILA dataset provides the manual segmentation in retinographies by two expert ophthalmologists, it is interesting to check whether the ISNT [29] rule is satisfied on this dataset. The dataset contains the retinographies and segmentations of both eyes of 161 healthy patients and 47 with glaucoma. For this aim, the rim width is calculated as the difference between the contour of the optic disc and the contour of the excavation. Then, the measures of each of the six sectors (TS, T, TI, NS, N, NI) are regrouped into four sectors, merging TS and NS into S and TI and NI into I. This is performed to assess compliance with the ISNT rule and its variants IST and IS [22] in patients in the PAPILA database [17,18]. The verification of the rules is tested both for each eye separately and in both eyes simultaneously. Table 2 gathers the results of the compliance of the ISNT, IST, and IS rule.

The results of Table 2 reveal that the greatest inconsistency in the percentage of compliance with each of the rules in both eyes occurs with the ISNT rule. Thus, among the three variants, it would be the most appropriate for a first approximation of the condition of the patient. Although this rule is fulfilled in more than double the percentage of cases in healthy patients than in glaucoma patients, only one-third of healthy patients comply with this rule. Therefore, non-compliance is not a sufficiently significant indicator on its own to determine the presence of the disease.

### 3.2. Relationship between RNFL Thickness in OCT and Rim Width in Retinographies

The next analysis is devoted to check any relationship between the value of the rim width in each sector extracted from retinographies and the RNFL thickness in the corresponding sector of the OCTs. Figure 4 shows these plots, using as representative the values for sectors T, N, TI, and G. In these plots, the linear regression model which better fits the cloud of points is indicated, as well as the RMSE (Root Mean Square Error) of the fit.

From Figure 4 it can be noted that there is no apparent distinction between the relationship in left or right eyes. In some sectors, the obtained point distribution is better fitted by a straight line as, for example, in the global sector (Figure 4a) or nasal inferior sector (Figure 4d), while in the temporal sector (Figure 4b) or nasal sector (Figure 4c), the points are arranged as a circular point cloud.

Regarding the distribution of glaucoma patients, most of them are concentrated in areas corresponding to lower thicknesses in both retinographies and OCTs, which is shown in the figures with a distribution generally closer to the origin compared to healthy patients. This is consistent with the fact that the increase in IOP leads to a widening in the optic cup with a consequent reduction in the thickness between the optic disc and optic cup (observed in retinographies) as well as a compression of the distances between the RNFL layer (thus reducing the thicknesses obtained in OCTs).

The value of the slope returned by the linear regression model equation tells us the relationship (on average) between the OCT values and the retinal images. This value is closer to unity in the G sector. We have used a scaling factor for the measurement of the thickness in the retinographies in order to obtain the greatest possible similarity between the average thickness of the G sector and that obtained in the G sector of the OCT scans. For this reason, the value of the slope is closer in the G sector of Figure 4. A factor of f=0.6425 has been applied directly to the measurements of all thicknesses, so that its effect appears in the graphs of all sectors.

With the aim of studying in more detail the relationship between the thickness of the fiber layer extracted from OCTs and retinographies, the values provided by both imaging modalities have also been analyzed using the asymmetry metrics defined in Section 2.3. In the study of the asymmetry characteristics’ relations, a procedure analogous to the prior analysis has been executed. The distribution of the data points suggests their use on their center of mass (or centroid) rather than applying a linear regression model. This study will concentrate exclusively on the overall thickness (G), as it provides the most significant conclusions. As shown in Figure 5, except for the asymmetry characteristics that employ the absolute value (|δ| and |Δ|), the relations between the results of the retinographies and the OCTs are presented as clouds of points with zero mean, as verified by observing the location of the centroid. The centroid corresponding to healthy patients is much closer to the origin of coordinates than in the case of patients with glaucoma, whose points appear much more dispersed and have a center of mass farther away. We note that this difference is especially significant in the metrics that use absolute values, suggesting that these characteristics may facilitate the differentiation between healthy and glaucoma patients.

### 3.3. Design of Classification Models Based on Features Extracted from Retinographies

#### 3.3.1. Classification Trees

The first proposal to design a classification model is a decision tree whose inputs are all the asymmetry metrics (Equations (Equation 2)–(Equation 7)) computed in all sectors (TS, T, TI, NS, N, NI, G). In this and all the studies performed, k−fold cross−validation with k = 5 splits is used. Initially, false positives and false negatives are equally penalized. The second column of Table 3 contains the results employing all the asymmetry features obtained from all sector thicknesses. From here on, these will be referred to as “All RET” or “All OCT”, according to its origin. The absolute values of the difference in CDR (|ΔCDR|) and in RDR (|ΔRDR|) between both eyes are also used as input characteristics in this column. With these features, patients with glaucoma were correctly classified in 44.7% of cases, while healthy patients were correctly classified in 78.3% of cases, leading to an accuracy of 70.7%. The better characterization of healthy patients compared to patients with glaucoma, as shown by higher specificity than sensitivity, is explained by the difference in the number of patients of each type (161 healthy patients versus 47 with glaucoma).

The next experiment is devoted to analyzing the most discriminative asymmetry feature. To that end, the inputs of the decision trees are the values of each asymmetry metric evaluated in each sector. The results of this comparison are also listed from the third column to the eighth column of Table 3. The asymmetry feature that provides the best classification tree is |Δ|, achieving a sensitivity of almost 50%, a specificity of 78.9%, and an accuracy of 72.1%. With $\bar{\bar{\Delta }}$, although the same accuracy is reached, a lower sensitivity value is obtained, which is a crucial parameter in the medical use of the study. Therefore, from now on, we focus on the |Δ| asymmetry metric. The ninth to eleventh columns of Table 3 show the results of adding the absolute difference in CDR and RDR between both eyes to this selected metric.

As can be seen, |ΔRDR| appears to be more effective than |ΔCDR| for patient characterization, given that better classification metrics are obtained for both healthy patients and those with glaucoma. We also note that including both parameters provides a worse classification capacity. It is then interesting to consider which of the previous characteristics are most relevant. These will be referred to as Most Significant Parameters, MSPs, to the most discriminant information. Figure 6 shows this result, obtained by averaging the relevance of each sector returned by the model in 1000 different decision trees. The most informative sectors for the algorithm are the TS, NS, NI, and mainly the global sector G. We also see a higher weight of the |ΔRDR| parameter over |ΔCDR|. The results of employing only these metrics are shown in Table 4.

Generally worse results are obtained when selecting only the best sectors of the |Δ| asymmetry metric compared to models with all sectors. Nevertheless, better results are also obtained by adding to these parameters the |ΔRDR| instead of the |ΔCDR|, although the metrics worsen compared to not using either of them. The results indicate that the worsening of the models when using more features is due to the “confusion” caused by certain features on the decision algorithm. In order to optimize the design of the decision trees, we conducted a study of its hyperparameter “Maximum Number of Splits” (MNS), which limits the growth of the tree to reduce the overfitting of the model to the training data. Figure 7 shows the evolution of the metrics obtained using |Δ| extracted from retinographies according to the maximum number of splits (MNS) allowed. In terms of accuracy, the minimum error is obtained when the maximum number of splits is restricted. Model performance worsens when the maximum number of splits is not limited, as the tree considers more and more patient features and tends to overfit to the training set values, thereby impairing its generalization ability and correct classification of new patients. This improvement is reflected in the characterization of healthy patients (specificity), increasing from 78.9% (in the non–restricted case, which coincides with the restriction of MNS = 20) to accuracy values above 93%. However, the correct classification of patients with glaucoma (sensitivity) is slightly reduced from 48.9% to values around 43%, which hardly deteriorates the most relevant metric of this study.

In order to increase the sensitivity, the costs assigned by the algorithm to patients with glaucoma misclassified as healthy have been modified (cost assigned to false negatives, C(FN)). This increases the penalization to false negatives while maintaining the cost of false positives to unity, C(FP)=1, thus improving the characterization of patients with glaucoma. Better sensitivities are obtained (increasing from 48.9% to 53.2%) when false negatives are penalized by an additional 20%. Changing this parameter corresponds to an increased ease for the algorithm to classify patients as having glaucoma, leading to an increase in the number of healthy patients mistakenly classified and reducing the specificity and precision of the model. This trend of increasing sensitivity and reducing specificity is not constant for any range of cost values, as for example, using C(FN) = 2, both metrics worsen compared to using C(FN) = 1.2, where an acceptable balance between the different metrics is achieved. Choosing too high of costs such as C(FN) = 4.2, which returns the maximum sensitivity of 61.7%, is not viable due to the considerable deterioration of specificity, which decreases from the initial 78.9% to 65.2%.

#### 3.3.2. Support Vector Machines

Support Vector Machines (SVMs) have also been considered in this work to classify healthy and glaucoma patients based on the asymmetry metrics. We have applied the SVM algorithm to the features extracted from retinal fundus images in order to compare the results with those obtained using decision trees. Table 5 shows the results of applying all the metrics of the retinal scan thicknesses as well as the |ΔCDR| and |ΔRDR|. As can be seen, the combination of asymmetry metrics with the |ΔCDR| and |ΔRDR| values slightly worsens the sensitivity of the SVM model. When adding the |ΔCDR| and |ΔRDR| separately, we see that, unlike the decision trees, using only the |ΔCDR| provides better values for both sensitivity and specificity.

Following a similar study to the one conducted with decision trees, the results of the SVM models obtained with the different asymmetry metrics are gathered in columns 5 to 9 of Table 5. Those from features $\bar{\Delta }$ and $\bar{\bar{\Delta }}$ are identical to those obtained with Δ. With the use of SVM, we observe that increasing the number of features extracted from retinal images improves the performance of the algorithm. This can be seen specifically with the different results obtained using the δ to |Δ| characteristics separately and when combined, as reflected in the metrics of Table 5, where the best evaluation is obtained by using all metrics simultaneously.

In terms of the cost associated to false negatives, a better sensitivity is achieved given that we penalize FNs to a greater extent, making it easier for more patients to be classified as having glaucoma. This affects the specificity of the model. To achieve an adequate tradeoff between both metrics, we could establish a sensitivity around 63% with an accuracy of 83% in the classification of healthy patients, as shown in Figure 8, which is higher than the results obtained from decision tree models using only retinal images characteristics (around 53% with specificities of 77% using C(FN)=1.2).

### 3.4. Design of Classification Models Based on Features Extracted from OCTs

#### 3.4.1. Decision Trees

Once the models with the asymmetry characteristics provided by retinographies have been studied, the results obtained using the asymmetry features given by the dataset of OCT images will be analyzed. Following a similar procedure, Table 6 shows the performance of decision trees using as input each one of the inputs separately as well as the combination of all metrics. The obtained models reveal a significant improvement compared to those solely based on retinal features, achieving 70.2% accuracy in correctly identifying glaucoma patients and 88.8% accuracy in identifying healthy patients using all metrics in all sectors for the most discriminative asymmetry metrics Δ and |Δ|. Note that the best classification tree for retinal features had a sensitivity of 53% and a specificity of 77% after feature selection and false negative cost modification.

In order to examine the impact of selecting the most relevant sectors of the best asymmetry metric, the relevance of each sector has been extracted by averaging 1000 decision trees. For Δ, the sectors TS, TI, NI, and G are selected to be the most significant parameter (MSP), while for |Δ| we will employ TS, TI, NS, and G, in order to consider in both cases some features of both the nasal and temporal sectors of the eye. The performance of the models obtained using only these features is presented in the last two columns of Table 6. Models obtained from the highest–weight sectors allow for improved patient classification, being the best decision trees up to this point, correctly classifying 72.3% of glaucoma patients and over 90% of healthy patients.

The results of the classification trees using characteristics extracted from OCTs have been analyzed, controlling the maximum depth of the trees (or maximum number of splits, MNS). The effect of this hyperparameter was then studied with the Δ and |Δ| asymmetry metrics. By sweeping the classification error in both models based on the MNS value, we obtained the results shown in Table 7. It is interesting to highlight the distinct behavior of the models generated from Δ and |Δ| when varying the maximum number of splits of the tree. Depending on the features used for their training, the model benefits or not by the modification of this hyperparameter.

Next, we analyze the improvement of sensitivity by modifying the penalty for false negative errors, C(FN). To do this, we study the variability of the metrics of the models generated from the MSPs of Δ and |Δ| in OCT according to the different costs assigned to FN. Given that false negatives are penalized more heavily, the algorithm classifies more patients as having glaucoma, causing an increase in healthy patients incorrectly classified and the subsequent decrease in specificity. There are intervals of C(FN) values that cause clear improvements in sensitivity. The models achieve a sensitivity of 80.9% using MSP of Δ from OCT with C(FN)=3.2, the highest obtained up to the moment with decision trees, as well as a specificity reaching 87% in the case of the model with MSP of |Δ| from OCT with C(FN)=2.2.

#### 3.4.2. Support Vector Machines

The results of using the characteristics extracted from OCT to train SVM−based algorithms are shown in Table 8. We will not indicate the results for all features since $\bar{\Delta }$ and $\bar{\bar{\Delta }}$ return the same results as Δ.

By using metrics extracted from OCTs, the models generated from |δ| and |Δ| stand out, achieving the best evaluation in terms of accuracy. Combining good metrics with others not so favorable in this case deteriorates the performance of the model compared to those generated using only the best features. In this case, by combining only the most favorable features, |δ| and |Δ|, we obtain the SVM model from OCT with the highest sensitivity (59.6%) and correct classification of 97.5% of healthy patients, with a final accuracy of 88.9%. This is the best obtained with SVM models along this study.

Next, we aim to improve that sensitivity by assigning different costs to the FN penalization. Given the high specificity, we can a priori afford to greatly increase C(FN). We have an acceptable margin of accuracy in characterizing healthy patients, and a slight reduction would be acceptable. The evolution of the metrics of the SVM models that use |δ| and |Δ| extracted from OCT with respect to this hyperparameter can be seen in Figure 9.

### 3.5. Design of Classification Models Based on Features Extracted from Retinographies and OCTs

#### 3.5.1. Decision Trees

Now we analyze whether the generated decision trees could yield better results by combining data extracted from both retinal and OCT scans. To this end, we study all asymmetry features from the thickness of both imaging techniques on one hand and, on the other hand, only the best asymmetry features (|Δ| from retinographies and Δ and |Δ| from OCT scans). We also check if adding the features of |ΔCDR| and |ΔRDR| and later only |ΔRDR| improves the results.

As can be seen in Table 9, a better model is obtained by using only the best asymmetry features from the retinographies and OCT, rather than all of them. Adding the |ΔCDR| and |ΔRDR| does not improve the result, which presents a sensitivity of about 70% and a specificity of 90.1%. If only the most relevant sectors of the best asymmetry features, corresponding to |Δ| from retinographies and Δ and |Δ| from OCT, are used, the results obtained improve slightly compared to selecting all the sections of the best parameters, correctly characterizing 70.2% of patients with glaucoma and 90.7% of healthy patients.

Once the best features have been selected, we now seek to improve the resulting metrics through the hyperparameter “maximum number of splits” (MNS). As seen in Figure 10, limiting the maximum number of divisions improves the classification of healthy patients, with accuracy rates above 90.7%, while it worsens the sensitivity of the model, which falls below the 70.2% obtained without limiting the growth of the tree.

Discarding the modification of the hyperparameter MNS, we focused on the effect of costs assigned to false negatives on the MSP tree of the best retinography and OCT features. The results are particularly favorable with costs around C(FN) = 1.8, providing a model with a sensitivity of 78.7% and a specificity of 87.6%. These metrics are very favorable, but their results are lower than those obtained from the optimal cost models using only the best sectors of the main characteristics extracted from OCT, with a sensitivity of 80.9% and a specificity above 85%.

Since the best decision tree model throughout the work has been obtained from the |Δ| MSP extracted from OCT with C(FN) = 2.2, a model optimization is applied by combining the modification of the cost with a sweep of the hyperparameter MNS. The evolution with respect to this factor is shown in Figure 11, which reveals a slight improvement in specificity when limiting the growth of the decision tree to 10 divisions, where the classification error is minimum. The best classification tree model obtained along the study is presented in Figure 12. The model has a sensitivity of 80.9% and a specificity of 88.2%.

#### 3.5.2. Support Vector Machines

We now consider the results provided by the SVM algorithm when combining data from both retinal imaging and OCTs. From previous results, we have found that the best SVM from retinal imaging was obtained by combining all its features, while for the OCT it was achieved with the metrics |δ| and |Δ|, so we also tried choosing only those parameters, obtaining the results in Table 10.

By combining all parameters from both retinography and OCT, a model is achieved with a sensitivity of 61.7% and a specificity of 88.8%. However, selecting only the best metrics from OCT allows an increase in specificity, reaching 95%. We will choose this last selection of parameters to verify its evolution with respect to the hyperparameter C(FN). We found a significant improvement in sensitivity to the unequal penalty between FP and FN, with a maximum sensitivity around C(FN) = 3.8. The SVM model generated using all retinography features and Δ and |Δ| from OCT with C(FN) = 3.8 represents the best SVM model obtained throughout the study, correctly characterizing 83% of patients with glaucoma and 81.4% of healthy patients. By comparing this SVM model with the best decision tree shown in Figure 12, which had a sensitivity of 80.9% and a specificity of 88.2%, we can confirm the robustness of the results obtained by the decision tree model compared to those returned by other machine learning algorithms. Then, this decision tree can be considered as a simple but effective method which offer a good tradeoff in classification of patients and simplicity, allowing its use in daily clinical practice for glaucoma screening. Finally, as a summary, Table 11 gathers the results provided by the main models studied throughout the work.

## 4. Discussion

Considering the asymmetry between both eyes in the thicknesses of the RNFL layer and the optical nerve head extracted using two different imaging modalities, the reliability and usefulness of classification models for the diagnosis of glaucoma can be compared. On one hand, models generated from measurements obtained through OCT produce generally better results when classifying healthy and glaucoma patients. On the other hand, models trained with data extracted from retinal images, although showing worse performance, can provide a first approach to the diagnosis, facilitating an initial screening since these devices are common due to the lower cost of the equipment when compared to OCT devices.

Analyzing the thicknesses in different sectors and the asymmetry metrics obtained through these two imaging modalities, a roughly linear relationship can be appreciated, but not precise enough to avoid one of the modalities. When comparing the asymmetry values, it can be observed that the cloud of points resulting from healthy patients spreads closer to zero than in the case of glaucoma patients. This indicates a relation between the results of both modalities and a different distribution for healthy and glaucoma patients, which could be interesting to investigate further in future studies.

The analysis of the ISNT, IST, and IS rules shows that the rule with the highest compliance in the dataset is the ISNT, despite the fact that only 30.43% of healthy patients and only 12.76% of glaucoma patients comply with it. Thus, the percentage of healthy patients is not high enough for this rule to provide an adequate indicator for the diagnosis of glaucoma.

Regarding the asymmetry metrics, those that generally allow for better classification and distinction between healthy and glaucoma patients have been Δ and |Δ|, which normalize the difference (with sign and in magnitude) of the thicknesses in each sector with respect to the sum of the thicknesses of that sector in both eyes of the patient. The absolute difference in CDR and RDR between both eyes of the patient has not provided particularly favorable results as asymmetry features.

Focusing on classification models, the choice of decision trees has been based on their simplicity and interpretability, also allowing us to know the most influential features in the classification process. A collection of decision tree models have been designed in order to obtain an adequate characterization of glaucoma patients which minimizes the number of false negatives. By applying a hyperparameter optimization process, we have obtained models which classify correctly 80.9% of patients with glaucoma and 88.2% of healthy patients. The general procedure applied during the work consisted in a first analysis of the features to be used and a later study and optimization of the hyperparameters. From the initial sensitivity values (focusing on decision trees) below 50% and specificity of 75%, these metrics have improved, both being above 80%. On the other hand, SVM algorithms have been applied to contrast and validate the results obtained by decision trees from both retinal and OCT images. Applying again a higher penalization for false negatives, we have achieved models with 83% sensitivity and 81.4% specificity.

## 5. Conclusions

In this work, the asymmetry between eyes has been analyzed as a method for early detection of glaucoma using retinal fundus images and optical coherence tomographies. The results derived from the morphological differences in the optic nerve and the peripapillary retinal nerve fiber layer of both eyes support the hypothesis that the asymmetry between both eyes is indicative of the presence of glaucoma and can be used for the diagnosis of the disease. Specifically, the width of the optic rim, the CDR and RDR from the retinal fundus images, and the thickness of the RNFL from the optical coherence tomography have been the measurements used as asymmetry features. The discrimination capabilities of these asymmetry features have been tested through classification trees and support vector machines, extending the research conducted in [14], where only the thickness of RNFL in OCTs and only decision trees were considered.

The results show that models based on asymmetry features derived from OCTs lead to better results than those trained with data extracted from retinal fundus images, as they provide models with a better ability to classify between healthy and glaucoma patients. The reason is that the asymmetry characteristics obtained from retinographies have a higher degree of ambiguity, even using manual segmentation by experts, which leads to lower reliability of these parameters compared to RNFL thickness obtained by OCT. Nevertheless, the information provided by retinographies is less expensive than that derived from OCTs, due to their lower cost and greater availability in medical centers. Therefore it can be used in a first differentiation of healthy patients from those who require more in−depth studies for an accurate diagnosis. Additionally, the results prove that sensitivity can be increased by using data collected jointly from the two medical imaging modalities.

Regarding the classification models, it has been verified that SVMs provide better classification results than decision trees. However, the simplicity of the trees, the transparency of their decision rules based only on thresholds, as well as the relatively acceptable results suggest that the use of trees should not be discarded in the disease screening process.

Future lines of work include the optimization of the models through parameter selection criteria, the combination of several of these ML methods by means of ensemble learning, or the improvement of those already used, for example, in the case of decision trees, by manipulating the criteria for the selection of predictors or the definition of impurity of nodes. Additionally, in order to deal with data from unbalanced classes, another improvement to be incorporated will be the use of SMOTE as a data balancing method. It would also be interesting to consider other characteristics of the asymmetry or to consider other ways of evaluating the models obtained, such as the F1 score, or the use of neural networks to contrast the results obtained with those obtained using deep learning techniques.

## Figures and Tables

**Figure 1 sensors-23-04737-f001:**
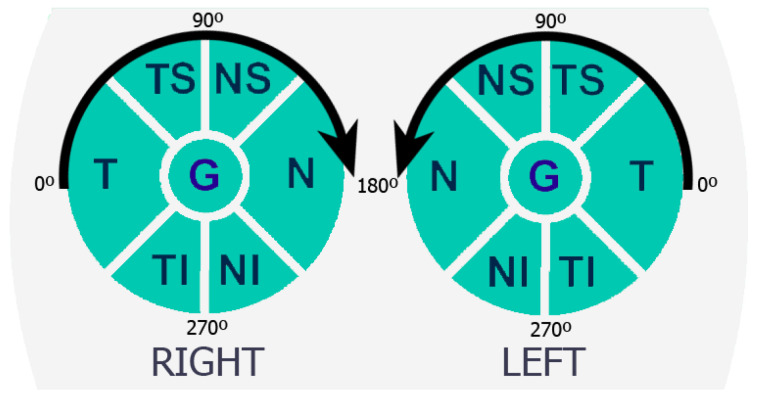
Distribution of the temporal (T), temporal superior (TS), nasal superior (NS), nasal (N), nasal inferior (NI), and temporal inferior (TI) sectors in both eyes of a subject.

**Figure 2 sensors-23-04737-f002:**
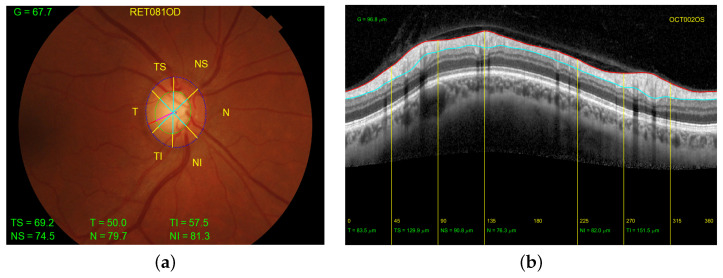
Distribution of the sectors. (**a**) Retinography of the right eye of a patient with the width of the rim from the manual segmentations of the optic disc (in dotted blue line) and excavation (in dotted green line). (**b**) Peripapillary B-scan OCT with the measurement of the thickness of the retinal nerve fiber layer delineated between the red line (top) and the light blue line (bottom). In both figures the yellow lines indicate the borders of each sector of the eye.

**Figure 3 sensors-23-04737-f003:**
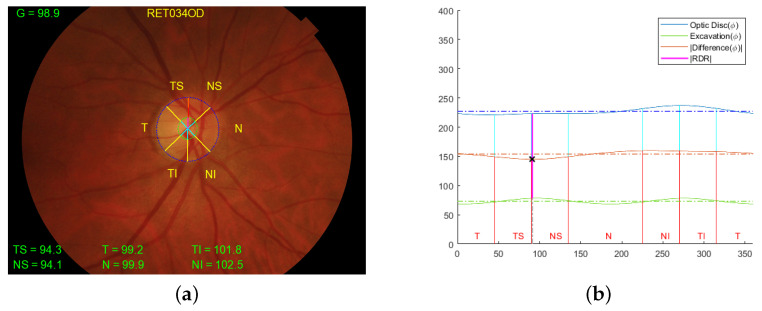
(**a**) Representation on retinal fundus images of the division by sectors and measurement of the thicknesses of the right eye of a healthy patient and (**b**) Cartesian coordinate representation of the RDR location and the division by sectors. The contour of the optic disc delineated in blue, the contour of the excavation in green, and the difference between the two in red. The horizontal lines show the mean values of each of these curves. The × mark indicates the angle of the RDR.

**Figure 4 sensors-23-04737-f004:**
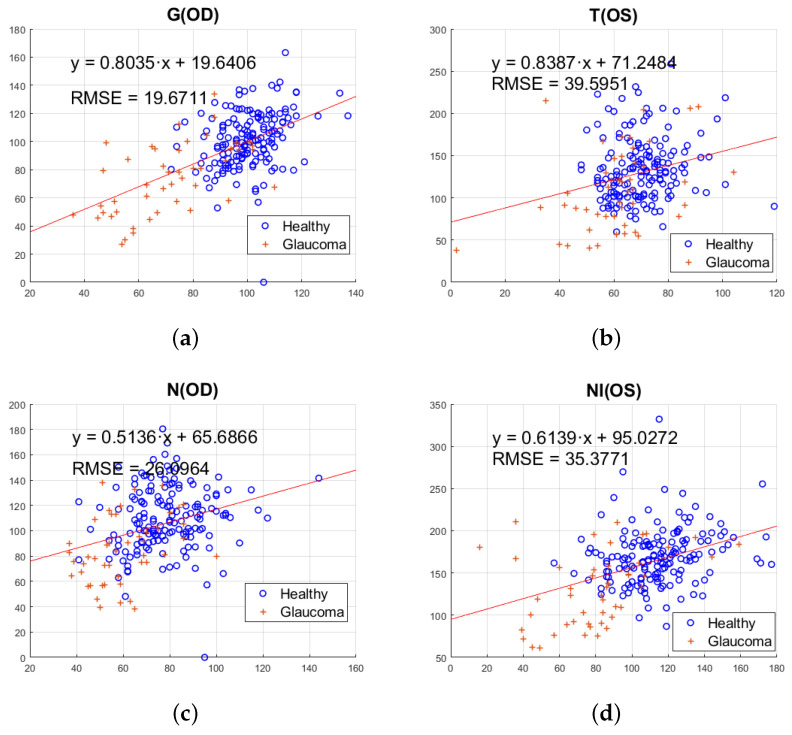
Thickness of the RNFL in OCT versus rim width in retinography considering right eyes (OD) or left eyes (OS) of healthy (’o’) and glaucoma (’+’) patients in (**a**) G sector, (**b**) T sector, (**c**) N sector, and (**d**) NI sector. The approximation by a linear regression model is also plotted.

**Figure 5 sensors-23-04737-f005:**
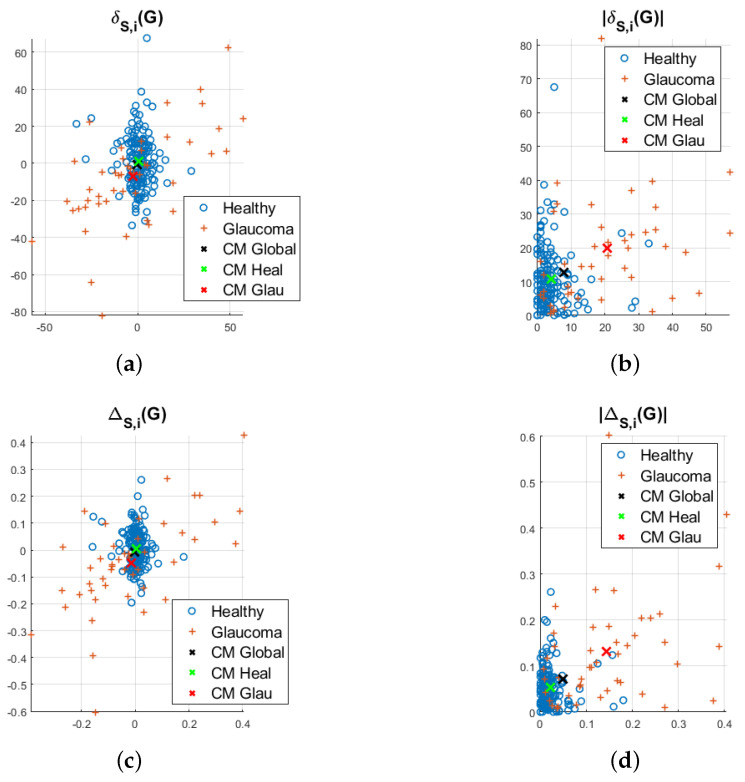
Relation of G−sector values obtained in RET versus OCT differentiating between healthy and glaucoma patients using the following asymmetry metrics: (**a**) δ, (**b**) |δ|, (**c**) Δ, and (**d**) |Δ|. The centroids (or centers of mass, CMs) are also depicted for all patients and each subset.

**Figure 6 sensors-23-04737-f006:**
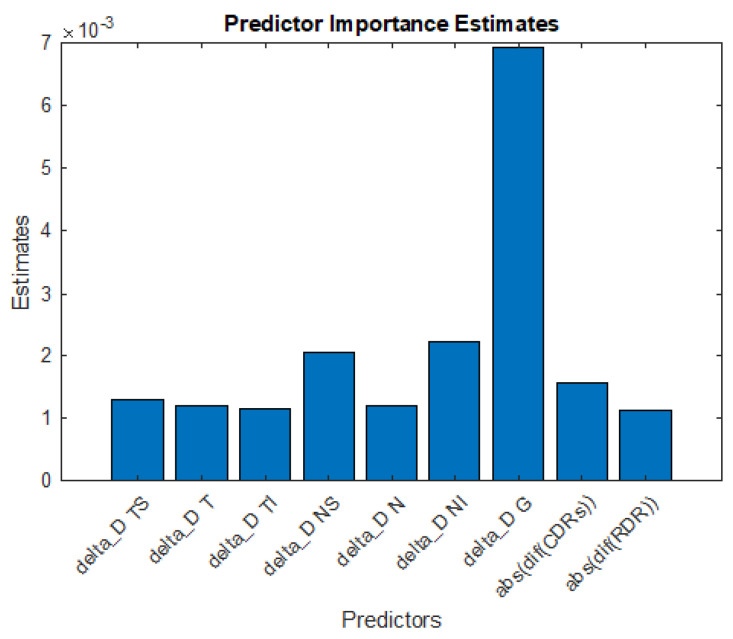
Relevance assigned by the decision tree algorithm to each sector of |Δ|, |ΔCDR|, and |ΔRDR|, extracted from retinographies.

**Figure 7 sensors-23-04737-f007:**
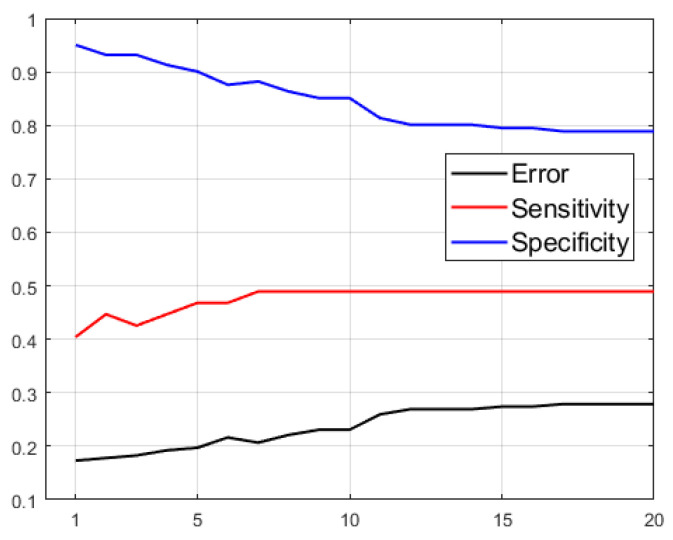
Evolution of the classification tree metrics as function of MNS trained with |Δ| extracted from retinographies.

**Figure 8 sensors-23-04737-f008:**
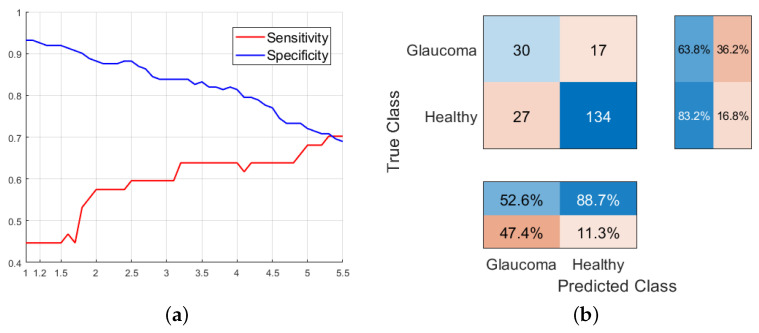
(**a**) Evolution of the sensitivity and specificity of the SVM model generated using all the metrics extracted from retinographies as a function of C(FN). (**b**) Confusion matrix of the SVM model using all the features from retinographies with C(FN)=3.5.

**Figure 9 sensors-23-04737-f009:**
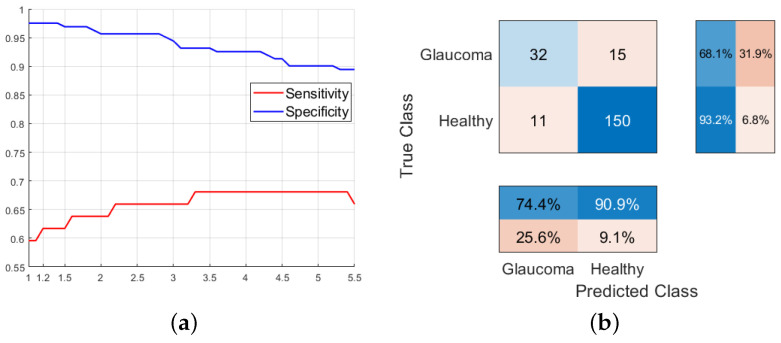
(**a**) Evolution of the sensitivity and specificity of the SVM model generated using |δ| and |Δ| extracted from OCT as a function of C(FN). (**b**) Confusion matrix particularized to C(FN)=3.5.

**Figure 10 sensors-23-04737-f010:**
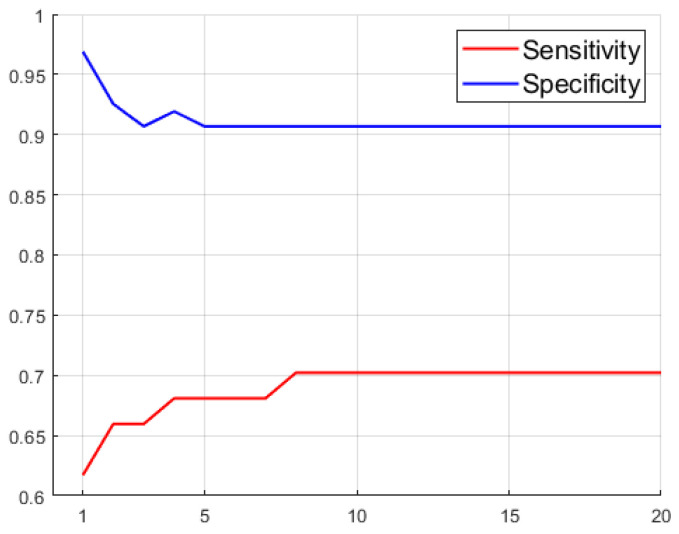
Evolution of the sensitivity and specificity of the decision tree trained with |Δ| of retinographies and Δ and |Δ| of OCT as a function of MNS.

**Figure 11 sensors-23-04737-f011:**
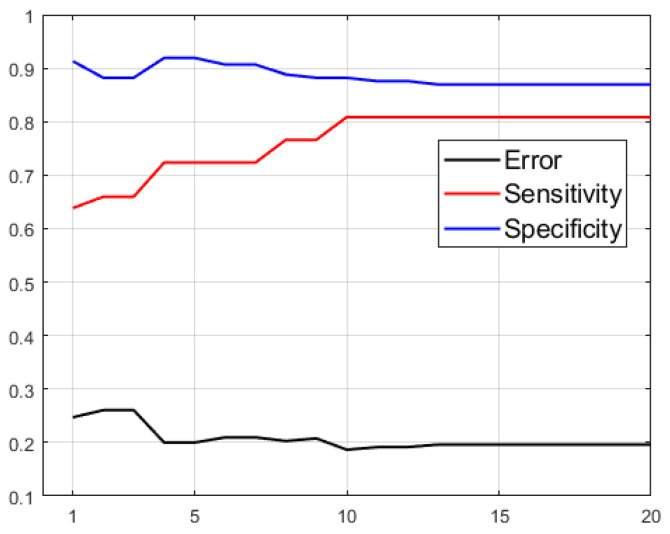
Evolution of decision tree metrics generated from |Δ| MSPs extracted from OCT with C(FN)=2.2 as a function of MNS.

**Figure 12 sensors-23-04737-f012:**
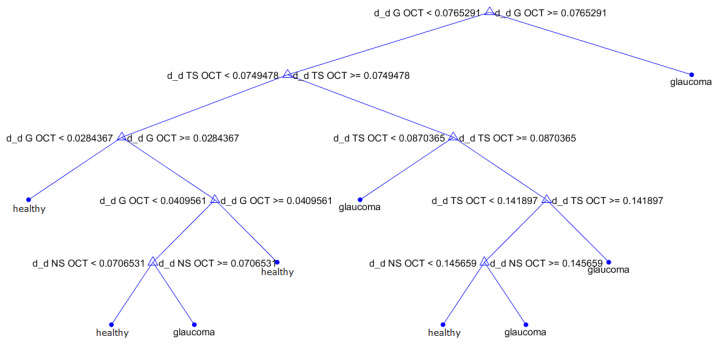
Optimal decision tree, generated from the |Δ| MSPs extracted from OCT with C(FN)=2.2 and MNS =10.

**Table 1 sensors-23-04737-t001:** Sex and age range of the healthy and glaucoma patients of the dataset.

	Healthy	Glaucoma	Total
Age (years)	59.10±12.70	70.15±9.21	61.61±12.85
Gender (male/female)	54/107	19/28	73/135
Total	161 patients	47 patients	208 patients

**Table 2 sensors-23-04737-t002:** Number and percentage of healthy patients (out of 161) and glaucoma patients (out of 47) of the PAPILA dataset complying with ISNT, IST, and IS rules in one eye (left or right) or both eyes simultaneously.

	Healthy (161 Patients)			Glaucoma (47 Patients)		
	ISNT	IST	IS	ISNT	IST	IS
Right eye	88 (54.66%)	97 (60.25%)	97 (60.25%)	16 (34.04%)	27 (57.44%)	27 (57.44%)
Left eye	71 (44.10%)	90 (55.90%)	90 (55.90%)	13 (27.66%)	25 (53.19%)	25 (53.19%)
Both eyes	49 (30.43%)	59 (36.65%)	59 (36.65%)	6 (12.76%)	17 (36.17%)	17 (36.17%)

**Table 3 sensors-23-04737-t003:** Performance of decision trees trained with asymmetry features extracted from retinographies.

	Input Asymmetry Characteristics									
	All RET + |Δ CDR| + |Δ RDR|	δ	|δ|	Δ	|Δ|	$\bar{\Delta }$	$\bar{\bar{\Delta }}$	|Δ| + |Δ CDR| + |Δ RDR|	|Δ| + |Δ CDR|	|Δ| + |Δ RDR|
**Sensitivity**	44.7%	40.4%	40.4%	29.8%	48.9%	40.4%	38.3%	44.7%	48.9%	51.1%
**Specificity**	78.3%	80.1%	78.9%	78.9%	78.9%	80.1%	82.0%	77.0%	74.5%	77.0%
**Precision**	37.5%	37.3%	35.8%	29.2%	40.4%	37.3%	38.3%	36.2%	35.9%	39.3%
**Accuracy**	70.7%	71.1%	70.2%	67.8%	72.1%	71.1%	72.1%	69.7%	68.75%	71.1%

**Table 4 sensors-23-04737-t004:** Performance of the models generated with the most significant parameter of |Δ|, MSP(|Δ|), and its combinations with |ΔCDR| and |ΔRDR|.

	Input Asymmetry Characteristics			
	MSP(|Δ|)	MSP(|Δ|) + |Δ CDR| + |Δ RDR|	MSP(|Δ|) + |Δ CDR|	MSP(|Δ|) + |Δ RDR|
**Sensitivity**	46.8%	40.4%	44.7%	46.8%
**Specificity**	78.3%	77.6%	70.0%	77.0%
**Precision**	38.6%	34.5%	36.2%	37.3%
**Accuracy**	71.1%	69.2%	69.7%	70.2%

**Table 5 sensors-23-04737-t005:** Performance of SVM models considering different sets of input features from retinographies.

	Input Asymmetry Characteristics								
	All RET + |ΔCDR| + |ΔRDR|	All RET	All RET + |ΔCDR|	All RET + |ΔRDR|	δ	|δ|	Δ	|Δ|	δ+|δ|+Δ+|Δ|
**Sensitivity**	44.7%	46.8%	46.8%	44.7%	29.8%	2.1%	0.0%	6.4%	44.7%
**Specificity**	93.2%	93.2%	93.2%	92.5%	96.3%	97.5%	100%	100%	92.5%
**Precision**	65.6%	66.7%	66.7%	63.6%	70.0%	20.0%	0.0%	100%	63.6%
**Accuracy**	82.2%	82.7%	82.7%	81.7%	81.3%	76.0%	77.4%	78.8%	81.7%

**Table 6 sensors-23-04737-t006:** Performance of decision trees considering the asymmetry metrics extracted from OCTs in all sectors and the most significant sectors of Δ and |Δ|.

	Input Asymmetry Characteristics								
	δ	|δ|	Δ	|Δ|	$\bar{\Delta }$	$\bar{\bar{\Delta }}$	All OCT	MSP Δ	MSP |Δ|
**Sensitivity**	48.9%	59.6%	68.1%	68.1%	48.9%	61.7%	70.2%	72.3%	72.3%
**Specificity**	93.2%	87.0%	91.9%	91.3%	93.2%	89.4%	88.8%	92.5%	90.7%
**Precision**	67.6%	57.1%	71.1%	69.6%	67.6%	63.0%	64.7%	73.9%	69.4%
**Accuracy**	83.2%	80.8%	86.5%	86.1%	83.2%	83.2%	84.6%	88.0%	86.5%

**Table 7 sensors-23-04737-t007:** Performance of decision trees based on OCT characteristics when modifying the maximum number of splits (MNS) with Δ and |Δ|.

	Input Asymmetry Characteristics			
	Δ, MNS = 20	Δ, MNS = 3	|Δ|, MNS = 20	|Δ|, MNS = 3
**Sensitivity**	68.1%	55.3%	68.1%	66.6%
**Specificity**	91.9%	94.4%	91.3%	95.0%
**Precision**	71.1%	74.3%	69.6%	79.5%
**Accuracy**	86.6%	85.6%	86.1%	88.5%

**Table 8 sensors-23-04737-t008:** Performance of SVM models trained with different asymmetry metrics using characteristics extracted from OCTs.

	Input Asymmetry Characteristics						
	δ	|δ|	Δ	|Δ|	$\bar{\Delta }$	δ+|δ|+Δ+|Δ|	All OCT
**Sensitivity**	2.1%	55.3%	0.0%	53.2%	0.0%	51.1%	51.1%
**Specificity**	100%	96.6%	100%	98.8%	100%	95.7%	96.3%
**Precision**	100%	83.9%	0.0%	92.6%	0.0%	77.4%	80.0%
**Accuracy**	77.9%	87.5%	77.4%	88.5%	77.4%	85.6%	86.0%

**Table 9 sensors-23-04737-t009:** Performance of classification trees generated with combinations of retinography and OCT features.

	Input Asymmetry Characteristics						
	All RET and All OCT	Best Metrics RET and OCT	Best Metrics + |Δ CDR| + |Δ RDR|	Best Metrics + |Δ RDR|	MSP Best Metrics RET and OCT	MSP Best Metrics RET and OCT + |Δ CDR| + |Δ RDR|	MSP Best Metrics RET and OCT + |Δ RDR|
**Sensitivity**	68.1%	68.1%	68.1%	68.1%	70.2%	68.1%	68.1%
**Specificity**	87.6%	90.1%	90.1%	90.1%	90.7%	90.1%	90.1%
**Precision**	61.5%	66.7%	66.7%	66.7%	68.8%	66.7%	66.7%
**Accuracy**	83.2%	85.1%	85.1%	85.1%	86.0%	85.1%	85.1%

**Table 10 sensors-23-04737-t010:** Performance of SVM models using characteristics extracted from retinographies and OCTs.

	Input Asymmetry Characteristics	
	All RET and All OCT	All RET and |δ|, |Δ| from OCT
**Sensitivity**	61.7%	61.7%
**Specificity**	88.8%	95.0%
**Precision**	61.7%	78.4%
**Accuracy**	82.7%	87.5%

**Table 11 sensors-23-04737-t011:** Summary table with the results provided by the main machine learning models designed throughout the work.

Method	Image Modality	Features Applied	Hyperparameter Modified	Sensitivity (%)	Specificity (%)	Precision (%)	Accuracy (%)
Decision trees	RET	All +|ΔCDR| +|ΔRDR|	−	44.7	78.3	37.5	70.7
		Δ	−	29.8	78.9	29.2	67.8
		|Δ|	−	48.9	78.9	40.4	72.1
		MSP |Δ|	−	46.8	78.3	38.6	71.1
		MSP |Δ| +|ΔCDR| +|ΔRDR|	−	40.4	77.6	34.5	69.2
		|Δ|	MNS = 3	43.5	93.0	−	81.0
			C(FN) = 4.2	61.7	65.2	34.1	64.4
	OCT	All	−	70.2	88.8	64.7	84.6
		Δ	−	68.1	91.9	71.1	86.5
		|Δ|	−	68.1	91.3	69.6	86.1
		MSP Δ	−	72.3	92.5	73.9	88.0
		MSP |Δ|	−	72.3	90.7	69.4	86.5
		MSP Δ	MNS = 3	55.3	94.4	74.3	85.6
		MSP |Δ|	MNS = 3	66.6	95.0	79.5	88.5
		MSP Δ	C(FN) = 3.2	80.9	85.1	61.3	84.1
		MSP |Δ|	C(FN) = 2.2	80.9	87.0	64.4	85.6
		MSP |Δ|	MNS = 10, C(FN) = 2.2	80.9	88.2	66.7	86.5
	RET + OCT	All	−	68.1	87.6	61.5	83.2
		MP	−	68.1	90.1	66.7	85.1
		MSP MP	−	70.2	90.7	68.8	86.0
		MSP MP	C(FN) = 1.8	78.7	87.6	64.9	85.6
SVM	RET	All	−	46.8	93.2	66.7	82.7
		All +|ΔCDR| +|ΔRDR|	−	44.7	93.2	65.6	82.2
		All	C(FN) = 3.5	63.8	83.2	52.6	78.8
	OCT	All	−	51.1	96.3	80.0	86.0
		|δ|,|Δ|	−	59.6	97.5	87.5	88.9
		|δ|,|Δ|	C(FN) = 3.5	68.1	93.2	74.4	87.5
	RET + OCT	All	−	61.7	88.8	61.7	82.7
		All RET, |δ|,|Δ| OCT	−	61.7	95.0	78.4	87.5
		All RET, |δ|,|Δ| OCT	C(FN) = 3.8	83.0	81.4	56.5	81.7

## Data Availability

The PAPILA dataset can be found at https://figshare.com/articles/dataset/PAPILA/14798004/1 (accessed on 20 April 2023). The OCT dataset is not publicly available at this moment.

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
