# Peer review of "Analysis of the Asymmetry between Both Eyes in Early Diagnosis of Glaucoma Combining Features Extracted from Retinal Images and OCTs into Classification Models"

_sensors, 2023, doi:10.3390/s23104737_

Round 1
Reviewer 1 Report
This study aims to analyze the asymmetry between both eyes of the same patient for the early diagnosis of glaucoma combining features extracted from retinal images and OCTs into classification models. It is interesting and innovative.
I have the following suggestions:
1. Models based on data extracted from retinographic images have not led to a good results , though the information provided by retinographic images is less expensive than that derived from OCT, It has limited clinical significance.
2. The result section is too redundant and cannot present the result clearly.
3. The discussion section could be more enriched.For example, the limitations can be clarified.
The language is good and smooth.
Author Response
Please find the responses to the review report in the attached file.

Reviewer 2 Report
The objective of this investigation is to examine the asymmetry between the patient's two eyes to aid in the early detection of glaucoma. Retinal fundus pictures and optical coherence tomographies (OCT) are two imaging modalities that have been taken into consideration for this aim. Spectral-domain optical coherence tomographies were used to assess the thickness of the retinal nerve fibre layer, while retinal fundus pictures were used to extract the cup/disc ratio difference. Good experiments are carried out, however, Before final publication, this work must address some revisions/concerns.
1. 1. What is novelty of the work. Please underscore the scientific value added/contributions of your paper in your abstract and introduction and address your debate shortly in the abstract.
2. A good article should include, (1) originality, new perspectives, or insights; (2) international interest; and (3) relevance for governance, policy, or practical perspective.
3. The work is devoted to an actual scientific and applied problem, performed by correct modern methods and the results are not in doubt. But the presentation and discussion of the results, as well as the conclusions, need to be improved.
4. Why two imaging modalities are analyzed for early diagnosis of glaucoma? Why single modality is not sufficient? Discuss pros and cons of both modalities.
5. Distribution of different classes seems imbalanced. Have authors utilized any data balancing methods such as SMOTE?
6. Discuss the notations used in equations 2 to 5.
7. Table 7. Performance of SVM models trained with specific asymmetry metrics extracted from retinographies. The results seems very poor event 0 % in table is provided. Check.
8. Figure 8. (a) Evolution of the sensitivity and specificity of the SVM model generated using all the
metrics extracted from retinographies as a function of C(FN). Check ‘ sensibility’ is mentioned in legend. It should be ‘sensitivity’. Why is sensitivity decreasing while specificity is increasing in graph? Justify.
9. Figure 10. Evolution according to the cost of the FN of the metrics of the classification tree model generated using: (a) OCT ∆ MSP. (b) OCT |∆| MSP. Why sensitivity and specificity are so much unstable?
10. Table 15. Summary table with the results provided by the main machine learning models designed throughout the work. What is ‘ Hiperparam’ ? is it hyperparameters?
11. Conclusion section is not provided. Add
Minor editing of English language required
Author Response

(The authors gave the same response as above.)

Reviewer 3 Report
Authors proposes to analyze the asymmetry between eyes for the early diagnosis of glaucoma. The analysis has been performed both on retinal fundus images and on OCT. They use SVM and decision trees to classify healthy and glaucoma patients.
I've found the paper very clear and well written. Every section is well described and the approach is well motivated. Nevertheless I'd suggest some minor change in order to improve the overall quality of the paper.
In section 2.1 I'd mention the patients age range and their sex.
Fig. 3 Yellow lines on white background result hardly readable.
Though accurate, section 3 is too long, maybe authors can shorten it by choosing to show only tables or graphs in some figure.
Fig. 17 is not needed
Author Response

(The authors gave the same response as above.)
